# The Presence of ANCA in IgA Crescentic Nephropathy Does Not Lead to Worse Prognosis with Intensive Rescue Treatment

**DOI:** 10.3390/jcm11237122

**Published:** 2022-11-30

**Authors:** Irene Agraz, Zaira Castañeda, María Teresa Sanz-Martínez, Alejandra Gabaldón, Sheila Bermejo, Laura Viñas Gimenez, Roxana Bury, Mónica Bolufer, Marina López-Martínez, Natalia Ramos, Oriol Bestard, María José Soler

**Affiliations:** 1Department of Nephrology, Vall d’Hebron University Hospital, Reference Center for Complex Glomerular Diseases (CSUR), 08035 Barcelona, Spain; 2Department of Immunology, Vall d’Hebron University Hospital, 08035 Barcelona, Spain; 3Department of Pathology, Vall d’Hebron University Hospital, 08035 Barcelona, Spain

**Keywords:** IgA nephropathy, antineutrophil cytoplasmic autoantibodies (ANCA), crescents

## Abstract

Background: Immunoglobulin A nephropathy (IgAN) is the most common glomerulonephritis worldwide. The concomitant presence of both crescentic proliferation and anti-neutrophil cytoplasmic autoantibodies (ANCA) in this pathology represents a rare coincidence. However, it is not clear to what extent the presence of ANCA (IgA or IgG) in these patients could have any clinical significance. The aim of the current work is to describe the presence of ANCA (IgA or IgG) in patients with IgAN and crescentic proliferation and its possible clinical implications. Methods: We retrospectively recruited all patients in our center with a histological diagnosis of IgAN with crescentic proliferation between January 2013 and December 2020. The main demographic and clinicopathologic data, fundamental histological characteristics, as well as the treatments implemented and main kidney outcomes, were collected and analyzed at a 6 and 12-month follow-up. Results: Between January 2013 and December 2020, a total of 17 adults were diagnosed with concomitant crescentic proliferation through a kidney biopsy of IgAN. Five (29.4%) patients showed ANCA, three (60%) showed IgA-ANCA and two (40%) showed IgG-ANCA. All ANCA-positive patients had some degree of crescentic proliferation. At diagnosis, the mean age of patients was 48 years old (range: 27–75). Nine of them were women (52%) and the most common clinical presentation was hypertension (71%). At the time of biopsy, the mean serum creatinine and proteinuria were 2.2 mg/dL (DS 1.42) and 3.5 g/mgCr (DS 1.22), respectively, with no statistical differences between ANCA-positive and -negative patients. Histological analyses showed that 11 out of the 12 (91%) ANCA-negative IgAN patients displayed less than 25% cellular crescents, whereas 100% of ANCA-positive IgAN patients displayed more than 25% cellular crescents (*p* = 0.04). Notably, five (30%) patients displayed fibrinoid necrosis, with four of them (80%) being IgAN-ANCA-positive (*p* = 0.01). Only one ANCA-negative patient needed renal replacement therapy (RRT) upon admission (5%). The mean serum creatinine and proteinuria were 1.94 mg/dL (DS 1.71) and 1.45 g/gCr (DS 1.78), respectively, within 6 months of immunosuppressive therapy. At 12-month follow-up, the mean creatinine was 1.57 mg/dL (DS 1). Four (23.5%) patients needed RRT at the end of the follow-up and four (23.5%) patients died. Conclusions: Probably due to the limited number of IgAN-ANCA-positive and IgAN-ANCA-negative patients, no significant differences were found between the clinical and laboratory characteristics. IgAN-ANCA-negative patients seemed to display less extracapillary proliferation than IgAN-ANCA-positive patients, who tended to show significantly higher fibrinoid necrosis. There were no differences regarding renal prognosis and patient survival after aggressive immunosuppressive therapy within 6 and 12 months when comparing the two samples.

## 1. Introduction

Immunoglobulin A nephropathy (IgAN) is the most common primary glomerulonephritis worldwide, with stable or slow progressive kidney disease occurring in most cases [1]. Nevertheless, some IgAN patients (<10%) have an acute or a subacute rapidly progressive course [2] due to the appearance of extracapillary proliferation with crescents unrelated to macroscopic hematuria, malignant hypertension or acute tubular necrosis [3]. The presence of crescent formation in biopsies with IgAN, could be considered predictors of renal evolution for this reason, its evaluation is considered important [4,5,6,7].

Furthermore, according to some authors, in a small proportion of patients (between 0.2 and 2%) with biopsy-proven IgAN, there are circulating anti-neutrophil cytoplasmic autoantibodies (ANCA), usually associated with the presence of extracapillary proliferation [4,5]. The association of ANCA and crescentic IgAN in patients is very rare and its significance has been poorly studied. In the majority of these patients, the type of antibody directed against MPO or PR3 is identified as IgG, but a few cases of patients with the positive IgA–ANCA type have been described [6,7].

The main aim of the current study was to analyze whether the presence of ANCA has any implications in the form of presentation, the histology, the evolution and the response to treatment in patients with positive IgAN-ANCA and crescents compared with those without ANCA.

We summarized the clinical manifestations, histological features, response to treatment and prognosis, and compared them between positive and negative IgAN-ANCA patients. We excluded patients with Henoch–Schönlein purpura, those with secondary glomerulonephritis, those who had undergone renal transplant and those with insufficient clinical and pathologic data.

## 2. Materials and Methods

A retrospective study was conducted between January 2013 and December 2020 at the nephrology department of the Vall d’Hebron University Hospital (HUVH) in Barcelona (Spain), to assess the effect of positive ANCA serology in IgAN patients with extracapillary proliferation, diagnosed via renal biopsy, on renal and patient prognosis. IgAN diagnosis was based on renal biopsy specimens from both ANCA-positive and ANCA-negative patients.

We conducted our study according to the oxford classification for IgA nephropathy, which establishes a score of mesangial proliferation (M), segmental glomerulosclerosis (S) and moderate-to-severe interstitial fibrosis and tubular atrophy (T), and adds the presence of any percentage of fibrous or cellular crescents (C), like in the study by Haas et al. [8]. Immunohistochemistry evaluated the presence of IgA, IgG, IgM, C3, C4, C1q and fibrin deposits (also graded on a scale of 0 to 3). Electron microscopy was performed when it was not possible to establish a diagnosis via light microscopy or immunofluorescence.

ANCA IgG or IgA were determined by indirect immunofluorescence using a BIOCHIP mosaic of ethanol (EOH)- and formaldehyde (HCHO)-fixed human granulocytes according to the manufacturer’s recommendations (Euroimmun, Lübeck, Germany). Secondary antibody FITC-labeled anti-human IgG or IgA (goat) was used (Euroimmun, Lübeck, Germany). Chemiluminescent immunoassay (CLIA) technology was employed to determine the autoantibodies nti-proteinase-3 (PR3) and/or anti-myeloperoxidase (MPO) (INOVA Diagnostics, San Diego, CA, USA). ANCA seropositivity was defined as being higher than 4.99 U/mL and 5.99 U/mL, respectively, in our laboratory.

Demographic and clinicopathologic data, subtypes, characteristics of the biopsy and treatment at the time of diagnosis were collected.

### Statistical Analysis

Categorical variables were expressed as proportions (%) and differences were assessed using the chi-squared test. Continuous variables were expressed as means ± standard deviation (SD) or medians (interquartile range). The differences between continuous variables were evaluated using the Student’s *t*-test and the categorical ones using the X^2^ and Fisher’s exact tests. The statistical package IBM SPSS V 23 (IBM, Armonk, NY, USA) was used. A *p*-Value of less than 0.05 was considered statistically significant.

## 3. Results

### 3.1. Clinical and Laboratory Characteristics

From January 2013 to December 2020, 17 adults were diagnosed with IgAN and extracapillary proliferation out of a total of 97 renal biopsies with IgAN diagnosis; this represented an incidence of 17.5%. Five of these patients presented with positive ANCA, three (17%) of them were IgA–ANCA patients and two (11%) were IgG MPO-ANCA patients, which resulted in an incidence of 5% of the total number of patients having IgAN and crescents. All of the positive ANCA patients had crescents. At diagnosis, the mean age of patients was 48 years old (27–75 years), and nine of them were women (52%). The demographics and clinical presentation of the two groups are represented in Table 1. The most common clinical presentation in both groups was hypertension.

The presence of edema, skin lesions and even systemic symptoms did not differ between the positive and negative ANCA groups. One negative ANCA patient needed replacement therapy at the time of diagnosis and did not recover renal function. Only one negative IgA–ANCA patient presented diffuse alveolar hemorrhage.

The laboratory data analysis showed that the median value for proteinuria was 3.5 g/mgCr (DS 1.22), for creatinine was 2.2 mg/dL (DS 1.42) and for hemoglobin was 11.7 mg/dL (DS 1.58). All patients had elevated microhematuria. Positive vs. negative ANCA characteristics did not differ between the two groups in either the clinical or laboratory data (Table 2).

### 3.2. Histological Characteristics

The tissue submitted to light microscopy consisted of the renal cortex with a number of glomeruli ranging from 1 to 33 (an average of 14.8 glomeruli). Glomeruli with cellular and fibrocellular crescent formation were observed in all cases. Of the examined glomeruli, the number with extracapillary hypercellularity ranged between 1 and 19 (an average of four crescents) (Figure 1). The presence of crescents was stratified according to proportions of crescents of less than 25%, between 26 and 50% and more than 50%, and we obtained the following results: eleven patients (64%) presented with extracapillary proliferation of less than 25% (all of them IgAN-ANCA-negative patients), and six patients presented with over 25% (four of them IgAN ANCA-positive patients). A total of five (30%) patients presented with fibrinoid necrosis in glomeruli, and four of them were ANCA-positive patients. The number of patients with renal biopsy depicting global esclerosis was higher in ANCA-negative patients, without statistical significance between two groups (Table 3). ANCA-positive patients did not differ in terms of tubular atrophy (40%) compared to ANCA-negative patients (37%). The same results were obtained with mesangial hypercellularity. No differences between the two groups were found in terms of histological characteristics except for ANCA-positive patients, who presented with significantly more fibrinoid necrosis than ANCA-negative patients (*p* = 0.01). Furthermore, ANCA-negative patients presented with less than 25% extracapillary proliferation in a higher proportion of renal biopsies compared to ANCA-positive patients (*p* = 0.04) (Table 3).

All cases showed dominant IgA staining in the mesangium via immunofluorescence.

### 3.3. Evolution and Response to Treatment

In terms of treatment, fifteen patients (88%), including all ANCA-positive IgAN patients, received intravenous steroids (pulse methylprednisolone, 7 mg/kg/day for 3 days) at the time of diagnosis, and thirteen patients (77%) also received cyclophosphamide (CP) (15 mg/kg, maximum 1.2 gr) for 6 months every 4 weeks if there were no complications. Only one ANCA-positive IgAN patient received rituximab (RTX) (two doses of 1 g), and two ANCA-negative IgAN patients received mycophenolate (MMF) due to contraindication with CP or medical decisions. We did not use plasmapheresis treatment in any case.

During follow-up, oral steroids were the main treatment at low doses (1 mg/kg/day), with progressive reduction until 5 mg/day was reached after 6 months, which was maintained for at least two years. MMF (2.0 g per day) was used as maintenance therapy in 15 patients for at least two years (Table 4). In one patient, we used azathioprine; in another patient, we used budesonide; and in another patient, we repeated RTX after 6 months of treatment (Table 4). At the time of diagnosis, one ANCA-negative IgAN patient, underwent renal replacement therapy without recovering kidney function.

At six months, the median creatinine was 2.2 mg/dL and the eGFR (estimated glomerular filtration rate) was 49.6 mL/min in ANCA-positive IgAN patients and 2.5 mg/dL and 48.2 mL/min for creatinine and eGFR, respectively, in ANCA-negative IgAN patients. There were no significant differences between the results of the two groups; this was maintained at twelve months and the trend seemed to be maintained throughout the follow-up (Figure 2).

There were no differences in terms of the need for renal replacement therapy or mortality according to ANCA presence; however, ANCA-positive patients seemed to have better evolution at the six-month follow-up, and this trend was maintained throughout the follow-up. At 12 months, good evolution was maintained, with mean creatinine of 1.63 ± 0.85 mg/dL in ANCA-positive IgAN patients and 1.62 ± 1.3 mg/dL in ANCA-negative patients, although the limited sample size must be taken into account (Figure 2). Four ANCA-positive IgA patients remained without renal replacement therapy requirements and only one patient died. Three more ANCA-negative IgAN patients required renal replacement therapy and one of them died; in addition, there were two more deaths in this group throughout the follow-up.

## 4. Discussion

In this retrospective study, we compare the clinical and histological features and response to treatment between two groups of patients with crescentic IgAN: ANCA-positive and ANCA-negative. This positive coincidence is very rare and has been scarcely described in the literature; it occurs in only 0.2–2% of IgAN patients [4,5], and its clinical implication are unknown. In our study, the incidence of ANCA-positive IgAN was higher than 5% in IgAN patients, and this may be, in part, related to the higher determination rate of IgA-ANCA in our center. On the other hand, the possibility of presenting ANCA-positive against IgA with negative IgG must be considered. This possibility is less prevalent but also described in the literature in undiagnosed patients with Henoch–Schönlein purpura [6,7], and we were also able to observe it in our patients. In our case, three of the five patients were positive IgA–ANCA patients. Some authors suggest that the presence of IgA antibodies is secondary to an increase in circulating immune complex cells of the IgA-fibronectin type [9]; others believe it is because the IgA-MPO reaction is mediated by abnormal composition in carbohydrates of the IgA2 molecule [10]; this was described in isolated cases of IgA nephropathy other than patients diagnosed with Henoch–Shönlein purpura, suggesting that these are clinical forms with a poor prognosis [6]. Due to the small sample size, we have not been able to establish differences between the subtypes of ANCA and we have only differentiated between ANCA-positive and -negative IgAN patients.

In our study, ANCA-positive IgAN patients showed a more severe histological picture in terms of fibrinoid necrosis and crescents; however, the renal prognosis is the same with more aggressive immunosuppressive therapy. Surprisingly, ANCA-positive IgAN patients seemed to have better renal outcomes in the short term regarding renal function, which was maintained throughout the follow-up with similar mean creatinine, even beyond 6 months, in the two groups; these results must be taken with caution due to the size of the sample. This is a finding that other authors had already previously described. Yang et al. [5] studied 20 ANCA-positive IgAN patients and found that they were older, had worse renal function, manifested more systemic symptoms including pulmonary involvement and, histologically, had a significantly higher percentage of fibrinoid necrosis in glomeruli compared to ANCA-negative patients. In our case, we did not find any differences in demographic or systemic symptoms. Pulmonary symptoms were not frequent in our case and there was only one ANCA-negative patient with lung involvement. Histologically, we concur with Yang et al. [5], in that our ANCA-positive patients had more fibrinoid necrosis than the ANCA-negative patients, who had a lower percentage of crescents. Regarding the number of crescents, it has been found that the presence less than 10% of crescents can lead to end-stage kidney disease in up to 40% of these patients [11], especially when crescents formation reaches 50%. To date, the mechanism of crescents formation in IgAN remains unknown. Recently Yu et al. [12] found that O-glycoforms of polymeric immunoglobulin A1 might contribute to crescents formation; they propose that the number of GalNAc of IgA 1 glycoforms was much lower in crescentic patients than in non-crescentic patients. However, the number of GalNAc was strongly associated with the percentage of crescents [12]. We consider that the presence of the minimum number of crescents in a renal sample is significant for the subsequent therapeutic approach, a fact that we have taken into account in our work and have included all those patients with the presence of any number of crescents.

Regarding the evolution, some authors, such as Bantis et al. [13], suggest that ANCA-positive IgAN patients represent a clinical entity with worse clinical and histologic presentation but not a worse response to treatment. Haas et al. [8], in concordance with Yang et al. [5], suggest that not all ANCA presence is pathogenic in these IgAN-positive patients, and recommend the identification of pathogenic versus non-pathogenic ANCA in these cases [14]. In our experience, ANCA-positive IgAN patients did not have more aggressive clinical presentation; however, they presented more fibrinoid necrosis and fibrous crescents. The presence or absence of ANCA in patients with IgAN and extracapillary proliferation in their renal biopsy was decisive in our case in starting early treatment with high doses of corticosteroids, based on the bolus of methyl prednisolone and cyclophosphamide, which seemed to have a good response in terms of renal survival even after the 6 and 12-month follow-up. Although other authors use similar treatments [13], there are no guidelines about how to treat this group of patients. The working subgroup of the IgAN Classification Working Group has addressed crescents as potential predictors of renal outcomes in IgAN but they do not give enough evidence to support the use of the Oxford Classification MEST-C score in determining whether immunosuppression should be indicated in these patients [11,15]. Recently, and due to the implication of the complement pathway in the pathogenesis of IgAN, it has been proposed that anti-complement therapies could be an alternative in aggressive cases or in those refractory to the proposed treatments [16,17].

In our study, four ANCA-positive IgAN patients had >25% of crescents and one of them had 100% of glomeruli with crescents; moreover, four of the ANCA-positive IgAN patients presented fibrinoid necrosis, confirming a more severe histological picture. However, these ANCA-positive patients did not differ in terms of tubular atrophy (40%) compared with ANCA-negative IgAN patients (37%). Despite this fact, in concordance with previous reports, evolution with intensive immunosuppressive treatment was positive in terms of renal survival in ANCA-positive IgAN patients [4,5], and currently, four of the five ANCA-positive patients do not require renal replacement therapy. In addition, in the majority of cases, corticosteroids and mycophenolate were used as maintenance treatments. This is similar to the treatment used in associated vasculitis, which was previously published [4,5,8] with good results in terms of renal prognosis in ANCA-positive IgAN.

The patients had no treatment-related complications during this follow-up period and most deaths were related to underlying respiratory and cardiovascular causes, with one having an oncological cause not related to the treatment of IgAN.

Finally, it is possible that the presence of ANCA-positive IgAN and crescents may be a genuine disease similar to the case of patients with antiglomerular basement membrane disease, and/or it may be that the coincidence of and IgAN and ANCA is a novel variant of crescentic IgAN [13,18]. Only multicenter studies with a large number of patients will be able to resolve this doubt.

There are several limitations to our study. One of them is the limited sample size related to the single-center study, and another is the short follow-up period. Moreover, there was a lack of protocols for the treatment of the induction and follow-up period due to the heterogeneity of the different studies reported, given that it is a rare pathology. Finally, the small number of patients did not allow us to establish major differences between ANCA IgA and IgG patients.

In conclusion, the unusual condition of ANCA-positive IgAN and crescents seems to have worse histological characteristics in terms of the number of crescents and the presence of fibrinoid necrosis; however, starting early and intensive immunosuppressive treatment may preserve renal function and improve the prognosis regardless of the type of ANCA at 6- and 12-month follow-ups, being the results no worse than those obtained in the cases of patients without ANCA and extracapillary proliferation. We recommend that doctors always be aware of the presence of any crescents in IgAN patients and that they screen, in these cases, for the presence of all types of ANCA, IgG and IgA. Given that the number of case series in patients with IgAN-ANCA and extracapillary proliferation are scarce, multicenter studies and clinical trials should be considered to establish its importance and treatment guidelines.

## Figures and Tables

**Figure 1 jcm-11-07122-f001:**
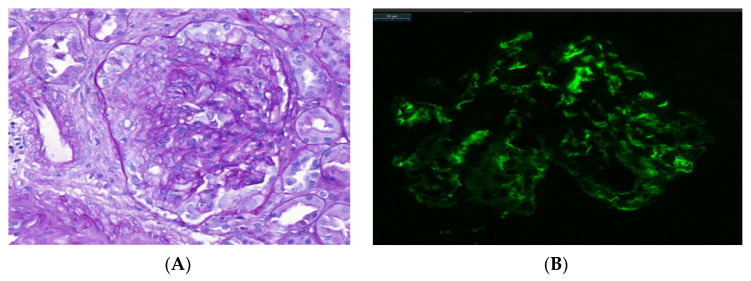
(**A**) Optical microscopy image, stained with PAS (Periodic Acid–Schiff). Increased endothelial–mesangial cellularity and extracapillary proliferation are observed in a patient with IgA nephropathy with crescents (original magnification ×20). (**B**) Immunofluorescence microscopy image shows intense (2+) mesangial IgA staining with granular pattern (bars: 20 μm).

**Figure 2 jcm-11-07122-f002:**
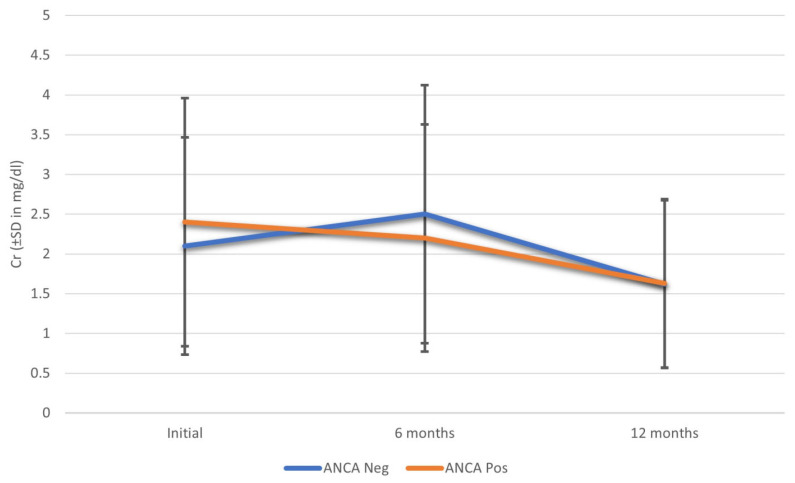
Renal evolution function with respect to creatinine in patients with IgA nephropathy and crescents according to the presence or absence of ANCA. Abbreviations: Cr—creatinine; SD—standard deviation.

**Table 1 jcm-11-07122-t001:** Clinical characteristics of patients with crescentic IgA nephropathy according to the presence or absence of ANCAs.

Patients	N = 17	ANCA-Positive N = 5 (3 IgA/2 IgG)	ANCA-Negative N = 12	* *p*-Value
Age (y)	48 (27–75)	54.6 (29–72)	46 (27–75)	0.51
Sex				
Women	9 (53%)	3 (60%)	6 (50%)	0.5
Men	8 (47%)	2 (40%)	6 (50%)	0.5
Clinical presentation				
Hypertension	12 (15%)	4 (67%)	8 (67%)	0.52
Edemas	7 (41%)	2 (40%)	5 (36%)	0.60
Skin lesions	4 (24%)	2 (40%)	2 (17%)	0.33
Lung involvement	1 (6%)	0	1 (8%)	0.71
Oligoanuria/Anuria	2 (12%)	1 (20%)	1 (8%)	0.51
Nephrotic syndrome	6 (35%)	2 (40%)	4 (33%)	0.6
Renal replacement therapy at diagnosis	1 (5%)	0	1 (8.3%)	0.7
Follow-up				
Renal replacement therapy dependence	4 (23.5%)	1 (20%)	3 (25%)	0.24
Mortality	4 (23.5%)	1 (20%)	3 (25%)	0.3

* *p*-Value less than 0.05.

**Table 2 jcm-11-07122-t002:** Laboratory data of patients with crescentic IgA nephropathy according to the presence or absence of ANCAs.

Laboratory	N = 17	ANCA-Positive N = 5 (3 IgA/2 IgG)	ANCA-Negative N = 12	*p*-Value
Hemoglobin (mg/dL) (DS)	11.7 (+/−1.58)	11.9 (+/−1.2)	12.1 (+/−1.26)	NS
Initial Creatinine (mg/dL) (DS)	2.2 (+/−1.42)	2.5 (+/−1.22)	2.1 (+/−1.26)	NS
Initial Proteinuria (g/mgCr) (DS)	3.17 (+/−3.5)	3.56 (+/−4.97)	3.2 (+/−3.04)	NS
Microscopic hematuria (HPF) *	505	661	439	NS
Creatinine (mg/dL) (DS) 6 months	2.25 (+/−1.8)	2.2 (+/−1.9)	2.46 (+/−2.1)	NS
Creatinine (mg/dL) (DS) 12 months	1.57 (+/−1)	1.63 (+/−0.85)	1.62 (+/−1.3)	NS

* HPF—high power field. NS—not significant.

**Table 3 jcm-11-07122-t003:** Histologic features of patients with crescentic IgA nephropathy according to the presence or absence of ANCAs.

	N = 17	ANCA-Positive N = 5 (3 IgA/2 IgG)	ANCA-Negative N = 12	*p*-Value
Glomerular				
Global glomerulosclerosis (%)	248	18.2	27.5	NS
Segmental glomerulosclerosis (%)	11	7.2	12.6	NS
Mesangial hypercellularity (%)	82	60	90	NS
Nº of crescents				
Crescents 0–25%	11 (64%)	0	11 (91%)	0.04
Crescents 26–50%	5 (29%)	4 (80%)	1 (8%)	NS
Crescents 51% o >	1 (6%)	1 (20%)	0	NS
Fibrinoid necrosis	5 (30%)	4 (80%)	1 (8%)	0.01
Tubule-Interstice				
Interstitial fibrosis/tubular atrophy				
Mild (<25%)	8 (47%)	3 (60%)	5 (40%)	0.04
Moderate (26–50%)	6 (35%)	1 (20%)	5 (40%)	NS
Severe (>50%)	3 (17.6%)	1 (20%)	2 (17%)	0.07
IgA staining (%)				
+	41.1	40	41.6	NA
++	13.8	40	16.6	
+++	35.3	20	41.6	

NS—not significant, NA—does not apply, + for mild, ++ for moderate, and +++ for strong.

**Table 4 jcm-11-07122-t004:** Immunosuppressive therapy in patients with crescentic IgA nephropathy according to the presence or absence of ANCAs.

	N = 17	ANCA-Positive N = 5 (3 IgA/2 IgG)	ANCA-Negative N = 12
Acute rescue treatment			
Corticosteroids	15 (88%)	5 (100%)	10 (83%)
Cyclophosphamide	13 (76%)	4 (80%)	9 (75%)
MMF	2 (12%)	0	2 (15%)
Rituximab	1 (6%)	1 (33%)	0
Maintenance treatment			
Corticosteroids	15 (88%)	5 (100%)	10 (83%)
MMF	15 (88%)	4 (80%)	11 (92%)
Budesonide	1 (6%)	0	1 (7%)
Rituximab	1 (6%)	0	1 (7%)
Azathioprine	1 (6%)	1 (33%)	0

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
