# Peer review of "The Presence of ANCA in IgA Crescentic Nephropathy Does Not Lead to Worse Prognosis with Intensive Rescue Treatment"

_jcm, 2022, doi:10.3390/jcm11237122_

Round 1
Reviewer 1 Report
Agraz et al. reports a series of patients with IgA nephropathy and crescents. Similar to prior studies, they found more crescents and fibrinoid necrosis in cases with ANCA seropositivity compared to those without. However, they did not detect difference in clinical and laboratory features.
I believe that the lack of difference in clinical and laboratory features between ANCA-positive vs. ANCA-negative cases are due to low sample size, as admitted by the authors. I believe this limitation should go into the abstract also.
I have following minor suggestions for revisions.
1) Please specify how IgA nephropathy diagnosis was made (Haas 2000 cited by the authors could be used as a reference).
2) Please provide intensity of IgA staining in the Table 3.
Author Response
Please see the attachment
Thanks a lot
I.Agraz

Reviewer 2 Report
The authors present an interesting work about the impact of ANCA on clinico-histological findings and prognosis in crescentic IgA nephropathy on a small case series. I have some comments and suggestions to improve the manuscript:
1. How many of your patients with IgA nephropathy (crescentic or not) are ANCA positive ?
2. It could be interesting to compare the prevalence of ANCA between proliferative and non-proliferative forms of IgA nephropathy.
3. In the results section "which resulted in an incidence of 5% of the total number of patients with IgAN and crescents". I think it is rather "5% of the total number of patients with IgAN"
4. The "Evolution and Response to treatment" paragraph is confusing and difficult to read. In the results section you state that "ANC-positive patients seem to have a better evolution", but you do not compare the kidney function data between the two groups. I think you should focus on comparing outcomes between ANCA+ and ANCA- IgAN patients.
5. Also, follow-up data about kidney function should be presented graphically in ANCA+ and ANCA- patients for better readability.
5. More extended follow-up would be appreciated and long term data about kidney function and need for dialysis would be interesting, as patients were included between 2013 and 2020.
6. Many typos and spelling mistakes are present in the text and tables and affect the readability of the manuscript: "immunosuppressive", "nephropathy", azathioprine", "cyclophosphamide", etc.. The manuscript must be re-read carefully and corrected.
7. Commas should be replaced by points between the numbers and the decimals in the tables.
8. CFM is not a commun acronym for cyclophosphamid, you might use "CP" instead
9. A figure of kidney pathology of ANCA+ proliferative IgAN would be appreciated to better illustrate your work.
Author Response
Please see teh attachment
Thanks a lot

Round 2
Reviewer 2 Report
I thank the authors for kindly addressing my remarks, and for adding important data. However I still have some remarks:
- The main problem for me is that the authors' conclusion is not consistent with their results.
Given their results, there is no difference in renal prognosis between ANCA+ and ANCA- patients in their cohort, not even a trend. The authors really should not state that this trend exists in the "Results" section, nor that there is a better renal prognosis in ANCA+ patients, and should temper their conclusions. In my opinion they should mostly conclude that ANCA+ patients do not seem to have a worse renal outcome.
Other remarks:
- One point is still unclear to me in the manuscript: is there 5 ANCA-positive patients among the global IgA cohort, or among crescentic IgA patients ? Please specify it more cleraly in the manuscript.
- Please add a scale bar or at least the magnification used in the microscopy images in Figure 1.
- Typos and English errors are still present through the manuscript and must be corrected.
